# Single versus Multiple Dose Ivermectin Regimen in Onchocerciasis-Infected Persons with Epilepsy Treated with Phenobarbital: A Randomized Clinical Trial in the Democratic Republic of Congo

**DOI:** 10.3390/pathogens9030205

**Published:** 2020-03-10

**Authors:** Michel Mandro, Joseph Nelson Siewe Fodjo, Alfred Dusabimana, Deby Mukendi, Steven Haesendonckx, Richard Lokonda, Swabra Nakato, Francoise Nyisi, Germain Abhafule, D Wonya’rossi, An Hotterbeekx, Steven Abrams, Robert Colebunders

**Affiliations:** 1Provincial Health Division Ituri, Ministry of Health, P.O. Box 57 Ituri, Congo; Michel.MandroNdahura@student.uantwerpen.be; 2Global Health Institute, University of Antwerp, 2610 Antwerp, Belgium; JosephNelson.SieweFodjo@uantwerpen.be (J.N.S.F.); Alfred.Dusabimana@uantwerpen.be (A.D.); Swabra.Nakato@uantwerpen.be (S.N.); an.hotterbeekx@uantwerpen.be (A.H.); Steven.Abrams@uantwerpen.be (S.A.); 3Centre Neuro-Psycho Pathologique, University of Kinshasa, P.O. Box 127 Kinshasa, Congo; debymukendi@yahoo.fr (D.M.); lokondarichard@gmail.com (R.L.); 4Interuniversity Institute for Biostatistics and statistical Bioinformatics, Hasselt University, 3590 Diepenbeek, Belgium; haesendonckx.steven@gmail.com; 5Centre de Recherche en Maladies Tropicales de l’Ituri, Rethy, P.O. Box 143 Bunia, Congo; fnyisi@gmail.com (F.N.); abhafule@gmail.com (G.A.); 6Programme National de Lutte contre l’Onchocercose, P.O. Box 185 Bunia, Congo; deogratiasrossy@yahoo.fr

**Keywords:** onchocerciasis, epilepsy, ivermectin, trial, seizures

## Abstract

Background: There is anecdotal evidence that ivermectin may decrease seizure frequency in *Onchocerca volvulus*-infected persons with epilepsy (PWE). Methods: In October 2017, a 12-month clinical trial was initiated in rural Democratic Republic of Congo. PWE with onchocerciasis-associated epilepsy experiencing ≥2 seizures/month were randomly allocated to receive, over a one-year period, ivermectin once or thrice (group 1), while other onchocerciasis-infected PWE (OIPWE) were randomized to ivermectin twice or thrice (group 2). All participants also received anti-epileptic drugs. Data was analyzed using multiple logistic regression. Results: We enrolled 197 participants. In an intent-to-treat analysis (data from group 1 and 2 combined), seizure freedom was more likely among OIPWE treated with ivermectin thrice (OR: 5.087, 95% CI: 1.378–19.749; *p* = 0.018) and twice (OR: 2.471, 95% CI: 0.944–6.769; *p* = 0.075) than in those treated once. Similarly, >50% seizure reduction was more likely among those treated with ivermectin twice (OR: 4.469, 95% CI: 1.250–16.620) and thrice (OR: 2.693, 95% CI: 1.077–6.998). Absence of microfilariae during the last 4 months increased the odds of seizure freedom (*p* = 0.027). Conclusions: Increasing the number of ivermectin treatments was found to suppress both microfilarial density and seizure frequency in OIPWE, suggesting that *O. volvulus* infection plays an etiological role in causing seizures.

## 1. Introduction

An association between onchocerciasis (river blindness) and epilepsy was reported as early as 1938 [1]. This association was later documented in many cross-sectional studies [2,3,4,5], but the causal relationship between onchocerciasis and epilepsy remains controversial. A recent cohort study performed in an onchocerciasis-endemic region in Cameroon strongly suggested that infection with *Onchocerca volvulus* is able to cause epilepsy depending on the microfilarial (mf) density [6]. 

To investigate the role of *O. volvulus* in triggering and aggravating seizures, we evaluated the effect of ivermectin on the frequency of seizures in onchocerciasis-infected persons with epilepsy (OIPWE). Demonstrating such an effect would provide additional support that infection with *O. volvulus* is able to cause epilepsy. Conducting a clinical trial in a remote onchocerciasis-endemic area in Africa is logistically difficult and costly to organize. Therefore, we initially performed a four months proof-of-concept trial to investigate the effect of ivermectin on the frequency of seizures in OIPWE treated with phenobarbital. The trial was performed in the Logo health zone, an onchocerciasis-endemic area with a high epilepsy prevalence (4.6%) in the Ituri province in the Democratic Republic of Congo (DRC), where ivermectin had never been distributed previously [7]. Between October and November 2017, a community-based treatment of epilepsy was initiated and 387 persons with confirmed epilepsy (PWE) were enrolled in the program. Ninety-four of them who met the criteria for onchocerciasis-associated epilepsy (OAE) [8], and who experienced at least two seizures per month by the time of assessment, were enrolled in a four-month proof-of-concept trial. A multiple logistic regression analysis showed a borderline insignificant association at 5% significance level between ivermectin treatment and being seizure-free at month 4 (OR: 1.652, 95% CI 0.975–2.799; *p* = 0.062) [9]. Given the small sample size, the results of the trial were difficult to interpret [9]. During the trial, we realized that the planned sample size (110 individuals) could not be reached as the inclusion criteria were very strict. Moreover, trial participants requested to be followed up beyond four months, and the PWE with *O. volvulus* infection who had initially been excluded from the trial also wanted to participate. Therefore, before completion of the proof-of-concept trial, we decided to re-randomize the initial 94 participants (group 1) into two arms: one arm would receive ivermectin once a year whereas individuals in the other arm would be treated thrice a year. Furthermore, we randomized the remaining 103 OIPWE who had initially been excluded from the trial (group 2) to ivermectin treatment twice or thrice a year. 

Ivermectin is very effective in killing the *O. volvulus* mf; however, it only temporarily represses mf production by the adult worm, which survives ivermectin treatment and resumes mf production at a slow rate after approximately 3–6 months [10]. Therefore, if ivermectin has an effect on the frequency of seizures in OIPWE, we expect that in a trial comparing annual ivermectin treatment with two or more doses per year, no difference in seizure frequency would be observed during the first few months. However, fewer seizures would be expected several months later particularly among persons who received more frequent doses of ivermectin. In this paper, we present the seizure outcomes of a single versus multiple dose ivermectin regimen in PWE after 12 months of follow-up. The CONSORT checklist and full protocol for this trial are available as Appendix A, respectively.

## 2. Material and Methods

### 2.1. Study Design and Participants

Before starting the recruitment of study participants, village chiefs, nurses, and community health workers (CHW) of five onchocerciasis-endemic villages (Draju, Kanga, Wala, Ulyeko and Thedeja) within the Logo health zone were informed about the purpose and specificities of the study. Study procedures, treatment regimens, potential risks and benefits were explained to interested participants and their parents/guardians. Consenting PWE were then assessed for eligibility to be enrolled into the study. Alur, the local language, was used for all communication with participants and parents/guardians.

To be enrolled, participants had to meet the 2014 International League Against Epilepsy (ILAE) definition of epilepsy: having experienced at least two seizures, unprovoked and without fever, with a minimum time difference of 24 hours between the two events [11] and either present mf detected during skin testing and/or onchocerciasis antibodies detected using an Ov16 rapid diagnostic test. 

Two skin snips were obtained from each participant (one skin snip from each iliac crest), using a Holth corneoscleral punch. After incubation for 24 hours in isotonic saline, mf were counted using an inverted microscope and the arithmetic mean of the mf in the two skin snips was calculated. The Ov16 rapid test (SD BIOLINE Onchocerciasis IgG4 rapid test, Abbott Standard Diagnostics, Inc. Yongin, Republic of Korea) was used to test for the presence of onchocerciasis IgG4 antibodies. 

### 2.2. Randomization and Treatment Regimens

Study participants were stratified into two groups; group 1 consisted of PWE who met the criteria for OAE [8] and experienced two or more seizures per month (these PWE were included in the four months proof-of-concept trial), and group 2 consisted of all other onchocerciasis-infected PWE (Figure 1). Separate randomization tables were developed for each group. Initially, the 94 PWE in group 1 were randomized to receive either 150 µg/kg ivermectin (Stromectol®, Den Haag, The Netherlands) [12] plus anti-epileptic drugs (AED) (phenobarbital), or AED alone. After four months of follow-up, these 94 PWE were re-randomized to receive a total of either one dose or three doses 150 µg/kg ivermectin (Stromectol®) plus AED. All the PWE in group 2 were randomized to receive 150 µg/kg ivermectin (Stromectol®) twice or thrice a year, plus AED during a one-year period. Overall, all study participants were followed for 12 months (M0 to M12) after the first dose of ivermectin.

Ivermectin (Stromectol®) was administered orally to the allocated study participants, under direct observation of an unblinded dispenser who was not involved in assessing the participants during follow-up. The time interval between two consecutive doses of ivermectin was always four months. All study staff involved in collecting and analyzing the data were kept blinded for treatment allocation until data lock. 

The AED (phenobarbital) dose was based on the participants’ weight: 5 mg/kg for participants weighing <15 kg; 3 mg/kg for those weighing between 15–35 kg, and 2 mg/kg for participants with a weight above 35 kg. Phenobarbital was taken orally once daily with possibilities to adjust the dose based on seizure frequency and/or occurrence of side effects. AED were made available freely with support of the humanitarian organization Malteser International. 

### 2.3. Baseline and Follow-Up Procedures

All study procedures were done according to standard operating procedures developed by the study team. At baseline, information was collected on seizure semiology, seizure frequency, epilepsy risk factors, relevant medical history, previous AED and ivermectin use. Women of childbearing age (14–49 years) were tested for HCG (human chorionic gonadotropin) in urine to exclude pregnancy, and pre-con ception counseling was given following national guidelines.

Trained community health workers (CHW) did monthly home visits, to monitor treatment adherence by counting AED pills, to identify potential side effects and to complete a seizure diary. Moreover, participants were seen monthly at the health center by project nurses and medical doctors during a one-year period. During visits, a physical and neurological examination was performed as well as an assessment of the frequency of seizures, adverse events, and adherence to AED treatment. Sub-optimal adherence was defined as ≥3 days per month without AED. Cognitive function was evaluated by determining whether the participant was well oriented in time and space, whether he/she could remember his/her name, was coherent in speech, and was obedient to orders. Medical doctors also reviewed the seizure diary and notes of the CHW. Based on their clinical assessment and information collected by CHW, medical doctors decided whether the initial dose of phenobarbital had to be continued or needed to be changed. If a participant was unable to reach the health center, a home visit was performed by a medical doctor or nurse trained in epilepsy management. Skin snip testing was repeated by the same laboratory technician every four months following the procedure described above. Skin snips were obtained before the administration of ivermectin. During each visit, female participants of childbearing age were questioned regarding menstrual aberrations and other signs of pregnancy. If pregnancy was suspected, a rapid pregnancy test was done and ivermectin treatment was interrupted upon confirmation of the gravid state but not the AED treatment. All pregnant women were followed up until delivery. 

### 2.4. Primary Outcome

The primary outcome was seizure freedom (all seizure types considered) between M9 and M12 of follow-up. Medical doctors compared the number of seizures reported in the diaries completed by the CHW with the number of seizures reported by the participants during each visit, and a consensual seizure frequency was reached between the doctor, the CHW, and the PWE/guardians. 

### 2.5. Secondary Outcomes

Secondary outcomes included: >50% seizure reduction between M9 and M12 of follow-up, skin mf density between M9 and M12 of follow-up, and adverse events at any period during the trial. Adverse events were assessed through clinical examination, by questioning participants and guardians, and by reviewing the reports of CHW. We used the Common Terminology Criteria for Adverse Events (CTCAE), version v5.0 for the classification of adverse events and the International Council for Harmonization of Technical Requirements for Pharmaceuticals for Human Use criteria to determine whether adverse events were serious [13].

### 2.6. Sample Size Calculation 

Assuming that 60% of OIPWE in group 1 receiving ivermectin thrice and 30% receiving ivermectin once will no longer have seizures during the last four months of follow-up (9 to 12 months after receiving the first dose of ivermectin) implying an odds ratio (OR) of 3.5, 84 (2 × 42) participants will be needed to detect a significant difference in odds between the treatment regimens with a power of 80% and a two-sided 5% significance level. Assuming that 65% of OIPWE in group 2 (most of them having less seizures compared to individuals in group 1) receiving ivermectin thrice will no longer have seizures during M9–M12 of follow-up, compared to 40% of those receiving ivermectin twice (OR = 2.5), 123 (2 × 62) participants will be required to detect a significant difference with a power of 80% and a two-sided 5% significance level. Assuming the rate of lost to follow-up or early withdrawal to be 10%, a minimum of 229 participants would be required (i.e., 94 in group 1, and 135 in group 2).

### 2.7. Statistical Analysis

All randomized participants who received at least one dose of ivermectin were included in the primary analysis according to their assigned treatment arms (intent-to-treat analysis). Categorical variables were reported as counts and percentages, while continuous variables were described using medians and interquartile ranges (IQRs). Baseline characteristics of the two groups (group 1 and 2) were compared using a Fisher’s exact test for categorical variables and a median test for continuous variables. Multiple logistic regression models were used to evaluate the effect of different ivermectin regimens on the probability of achieving seizure freedom or >50% seizure reduction during M9–M12 of follow-up. The relationship between seizure freedom during M9–M12 and the presence of mf in skin snips during the last four months was evaluated using multiple logistic regression. Stratified analyses by group were performed and model fit was compared with a combined model fitted to all data using the Akaike Information Criterion (AIC) [14]. An additional analysis based on the actual number of ivermectin treatments received (as-treated) was also performed. All PWE who died, were lost to follow-up, or discontinued their participation before the 12th month were considered as not being seizure-free. In the presence of separation, Firth’s modified score equation approach was used to estimate the model parameters [15]. The seizure frequency at continuous scale was modelled as a mixture of two separate distributions: first, a Poisson distribution to model the observed seizure frequency among PWE who had at least one seizure during the last four months; and second, a logistic regression to model the probability of seizure freedom during M9–M12 of follow-up. The performance of this model was compared with Zero-inflated Negative-Binomial regression model using AIC. The detailed output from the Zero-inflated Poisson regression model is provided in the Appendix A. The 95% confidence intervals (CIs) for all estimates were constructed. All analyses were performed using the statistical software programs SAS 9.4 (SAS Institute Inc.) and R version 3.6.1, relying on a two-sided 5% significance level. 

### 2.8. Ethical Considerations

The study was approved by the Ethics committee of the School of Public Health of the University of Kinshasa in the DRC (ESP/CE/063/2017) and the one of the University of Antwerp, Belgium (17/32/369). All participants gave their consent or assent (with parental consent) through signature or thumbprint in the presence of a literate witness. PWE who refused to participate were still examined and given AED. In specific cases, special consent was obtained for photos. 

### 2.9. Trial Registration

The clinical trial was registered at www.clinicaltrials.gov (NCT03052998) 

## 3. Results

### 3.1. Description of the Study Population

Between October 1st 2017, and March 16th 2018, 387 PWE were screened. One hundred ninety-seven PWE (50.9%) with evidence of *O. volvulus* infection were included in the trial. Ninety-four of them fulfilled the criteria for OAE with ≥2 seizures/months (group 1), of which 45 (47.9%) were randomly allocated to receive ivermectin once and 49 (52.1%) to receive ivermectin thrice. Of the remaining 103 OIPWE (group 2), 52 (50.5%) were randomly allocated to receive ivermectin twice and 51 (49.5%) to receive ivermectin thrice (Figure 2). Overall, 100 participants were assigned to receive ivermectin thrice, 52 twice, and 45 once. 

Participants from group 1 were significantly younger, with a higher prevalence of burn scars and cognitive impairment compared to those in group 2 (Table 1). 

A total of 163 (82.7%) participants attended their M12 follow-up visit; 18 (9.1%) were lost to follow-up, four (2.0%) withdrew their consent, and 12 (6.1%) had died (Table 2). Although the median mf density at month 12 was similar in both study groups, a greater proportion of participants in group 1 still experienced seizures and itching by the end of the study period compared to individuals in group 2. Details of the exact seizure types experienced by study participants at M0 and M12 are provided in the Appendix A. 

Of the 100 participants randomized to ivermectin thrice, 77 (77%) were treated as randomized. The remaining 23 did not receive all three doses of ivermectin: five of them died, five were pregnant, 12 were lost to follow-up (missed ivermectin dose because they did not come to the health center), and one withdrew consent. Of the 52 participants who were randomized to receive ivermectin twice, 45 (86.5%) attended the last follow-up visit; three died, two were lost to follow up, and two withdrew consent. However, only 43 participants in this arm received the two doses of ivermectin as randomized because 2 of them became pregnant. Of the 45 participants who were randomized to receive ivermectin once, 41 (80%) showed up for ivermectin treatment at M0 (four lost to follow-up) and one pregnancy occurred during the follow-up, although the pregnant participant had already received her ivermectin dose. In this single dose arm, 36 participants were followed up till M12 (four died, one withdrew consent).

Group 1 included a higher number of PWE for which the AED dose increased above 30 mg during the trial (Table 2). Moreover, the >30 mg increase in AED dose was slightly more frequent in persons on an ivermectin once regimen (in 21 out of 45 persons, 46.6%) compared to ivermectin twice (in 14 out of 52 persons, 26.9%) and thrice regimen (in 30 out of 100 persons, 30%). In group 1, ivermectin thrice implied higher odds of experiencing >30 mg increase in AED dose compared to ivermectin once (unadjusted OR: 2.142, 1.034–4.457; *p* = 0.039); no such difference was observed in group 2 between individuals on ivermectin twice and ivermectin thrice regimens (*p* = 0.786). 

### 3.2. Intent-to-Treat Analysis

The intent-to-treat population included all 197 randomized participants. Of these, 28 did not take the total number of ivermectin doses as assigned. In group 1, six of the participants assigned to thrice ivermectin were not treated as randomized (four treated once and two treated twice). In the ivermectin thrice arm of group 2, five participants were treated once and seven treated twice, giving a total of twelve not treated as randomized. Moreover, four PWE withdrew their consents, eight died before completing the assigned ivermectin regimen, and seven pregnancies occurred during the follow-up period. Regarding AED treatment, nine participants were switched to another AED, and 65 participants experienced >30 mg dose increase during the follow-up period. Additionally, 73 (37.1%) PWE did not adhere optimally to AED. On an intent-to-treat basis, 43% of participants were seizure-free during the last four months (M9 to M12) of the follow-up period. 

#### 3.2.1. Primary Outcome: Probability of Seizure Freedom during the Last Four Months

(1) Group 1 (OIPWE Meeting the OAE Criteria with ≥2 Seizures/Month at Baseline)

In group 1, the probability of being seizure-free during M9 to M12 was not significantly different in participants who received ivermectin thrice compared to those treated once (*p* = 0.111) (Table 3).

(2) Group 2 (OIPWE not Meeting the OAE Criteria or Experiencing <2 Seizures/Month) 

In group 2 the probability of being seizure-free during the last four months for PWE treated with ivermectin thrice was not significantly different than for those treated twice (*p* = 0.107) (Table 4). 

(3) Group 1 and 2 Combined 

Adjusting for study group membership (group 1 and group 2), multiple dose ivermectin regimen resulted in a higher odds of seizure freedom during the last four months of follow-up as compared to a single dose ivermectin regimen. After adjusting for other covariates, the odds of seizure freedom were 5.087 (95% CI: 1.378–19.749; *p* = 0.018) times higher for ivermectin twice as compared to the odds for a single ivermectin dose, and 2.471 (95% CI: 0.944–6.769, *p*= 0.075) times higher for ivermectin thrice as compared to a single dose (Table 5). Note that the comparison of treatment arms within each study group resulted in an OR of 2.471 (95% CI: 0.944–6.769) for ivermectin thrice versus once in group 1, and 0.486 (95% CI: 0.199–1.184) for ivermectin thrice versus twice in group 2, thereby explicitly acknowledging potential differences between the groups as the model is completely saturated for treatment regimen and group combinations. We also observed that >30 mg AED dose increase was strongly associated with reduced odds of seizure freedom (*p* < 0.001). The combined model presented in Table 5 outperforms a model allowing for differential effects of the covariates for both study groups (as is the case for the separate analyses) based on a smaller AIC-value (183.03 versus 193.79). 

#### 3.2.2. Secondary Outcomes

(1) Probability of >50% seizure reduction during M9 and M12 of follow-up

Adjusting for study group membership (group 1 and group 2), participants who received ivermectin twice or thrice were more likely to achieve >50% seizure reduction during M9–M12 as compared to those on a single dose ivermectin regimen (Table 6). 

(2) Skin microfilarial density o participants during follow-up

In both groups, the mean skin mf density decreased drastically between M0 and M4, and remained low at M8 through M12 (Figure 3). In group 1, the median mf density at M8 was similar in participants who received ivermectin thrice and those who were treated only once (*p* = 0.244). In contrast, at M12, the median mf density was significantly lower (*p* = 0.032) in participants who received ivermectin thrice compared to those who received ivermectin once (Figure 3A). A similar observation was made among individuals in group 2, with the difference in mf density between those treated thrice and twice being significant only at M12, but not at M8 (Figure 3B). 

The probability of being seizure-free was positively associated with the absence of mf during M9-M12; OR = 2.618 (95% CI: 1.136–6.289; *p* = 0.027) (Table 7).

### 3.3. As-Treated Analysis 

#### 3.3.1. Group 1

In group 1, of the 49 randomized to receive ivermectin thrice and the 45 randomized to receive ivermectin once, 39 and 41, respectively, were treated as randomized (Table 2). The ivermectin thrice regimen was associated with higher odds of seizure freedom compared to a single dose regimen (OR = 3.318; 95% CI: 1.161–10.500; *p* = 0.035) (Table 8). 

#### 3.3.2. Group 2

In group 2, 37 participants in the ivermectin thrice arm and 49 participants in the ivermectin twice arm were treated as randomized (Table 2). The probability of seizure freedom of the participants who received ivermectin thrice and those who received ivermectin twice was similar (*p* = 0.183) (Table 9). 

#### 3.3.3. Group 1 and 2 Combined 

In total, 77 participants received ivermectin thrice, 58 ivermectin twice and 50 ivermectin once. The participants who received ivermectin thrice and twice had higher odds of achieving seizure freedom compared to those on a single dose regimen; odds ratios of 4.795 (95% CI: 1.790–14.089) and 10.033 (95% CI: 2.670–42.496), respectively (Table 10). 

### 3.4. Trial Adverse Events

In total, 169 adverse events were reported among the 197 randomized participants; 133 (79%) of them were considered to be possibly related to ivermectin or the AED (Table 11). 

One serious adverse event, toxic epidermal necrosis (TEN), was related to the intake of phenobarbital. The affected person was a 42 years old male weighing 46 kg and randomized in the ivermectin thrice arm. He received a first dose of ivermectin at the start of the study and was treated with 100 mg of phenobarbital daily, indicated for generalized tonic-clonic seizures (one seizure per month at the start of the study). 

Two weeks after starting phenobarbital he experienced itching. A maculo-papular skin rash was noted at the first month visit, predominantly on the limbs. Ten days later, he was hospitalized at the Logo hospital because he was severely ill, presenting with fever, injected conjunctiva, and swollen lips with oral ulcerations. On day 6 of hospitalization, he developed bullous lesions over his entire body except his feet (Figure 4). A diagnosis of toxic epidermal necrolysis (TEN) was made; phenobarbital was stopped, and treatment was switched to sodium valproate. He was treated in the intensive care unit with corticosteroids and antibiotics for two weeks. Thirty days after admission, his skin lesions had healed completely. 

Five participants developed burns; four in the ivermectin thrice arm and one in the ivermectin once arm. In one of them, the burn was classified as a life-threatening serious adverse event requiring hospitalization.

Five participants died in the ivermectin thrice arm, three in the ivermectin twice arm, and four in the ivermectin once arm for reasons not directly related to study procedures or drugs. More specifically, four participants died during seizures (two of them by drowning in the river and two during seizures while sleeping), seven died of co-morbidities as detailed below: hospitalized and diagnosed with infection, malnutrition and anemia (2); septicemia (1); suspected pulmonary tuberculosis (1); severely altered consciousness (2); Nakalanga features and malnutrition (1). One patient who was lost to follow-up also died of an unknown cause; the death was reported by the family.

Eight women got pregnant during the study period. One of them, 32 years old, randomized in the ivermectin twice arm and taking daily 100 mg of phenobarbital since November 2017, delivered a girl in June 2019 with a separation of the upper lip not extended to the base of the nose and not involving the palate. The birth weight of the newborn was 3.5 kg and there were no other visible malformations. The Venereal Disease Research Laboratory test for syphilis of the mother was negative. In February 2018, this woman had received four tablets of praziquantel and during pregnancy she received two doses of sulfadoxine-pyrimethamine and 5 mg folic acid daily for three months. Since AED initiation in November 2017, she had been seizure-free. 

## 4. Discussion

In the intent-to-treat analysis of all OIPWE participants enrolled in the study (group 1 and 2 combined), those receiving multiple doses of ivermectin were more likely to be seizure-free during the last four months of the trial, compared to OIPWE who only received ivermectin once. Analyzing the results of the two study groups separately, an advantage of the thrice ivermectin regimen over the once or twice ivermectin regimen could not be confirmed. However, this may be because of the small sample size and because a large percentage of persons randomized to receive ivermectin thrice did not receive all doses, respectively 20.4% in group 1 and 27.5% in group 2. The advantage of the multiple dose regimens was shown in the combined as-treated analysis of the two groups, but also in analyzing the group 1 data alone. 

Ivermectin interacts with the glutamate-gated chloride channel receptors (GluClRs) activity that mediate neuronal and muscular inhibition [16]. However, GluCIRs are expressed exclusively in invertebrates [16]. In the mammalian central nervous system (CNS), ivermectin interacts with at least three targets: a GABA-dependent chloride channel, a glycine-dependent chloride channel and a voltage-dependent channel [17,18]. Through these interactions ivermectin may cause a dose-dependent CNS depression (including seizure suppression) in rats [18,19]. However, the penetration of ivermectin into murine CNS must be interpreted in the light of evidence demonstrating an increased permeability of their blood-brain barrier (BBB) during the neonatal period [20]. Ivermectin given at therapeutic doses is unlikely to cross the human BBB [21] and therefore cannot elicit a direct anti-epileptic effect in the CNS The relative superiority of the multiple dose ivermectin in reducing seizures during our study is probably explained by the lower mf density at the end of the study period in participants on this regimen, thus confirming the previously documented positive association between seizure frequency and mf density [22]. A small study conducted in a non-onchocerciasis-endemic area reported that treatment with ivermectin decreased the frequency of seizures in persons with refractory epilepsy [23]. However, in that study, ivermectin was given frequently (10 mg/day) and potentially interacted with the AED metabolism of the enrolled patients. Also, baseline seizure frequency was highly variable, and some patients had brain lesions which possibly compromised the integrity of the blood-brain barrier. All these elements make it difficult to interpret the results of that study.

Our study illustrates that epilepsy in Africa is associated with high mortality. Over a period of one year, 12 (6.1%) of the 197 OIPWE enrolled in the trial died, despite the availability of free AED. This number could be even higher, as 18 participants were lost to follow-up. A high mortality due to epilepsy, between 200 and 300 per 100,000 person-years was documented in onchocerciasis-endemic villages in Maridi, South Sudan [24]. Also, in a cohort study in an onchocerciasis-endemic area in Cameroon, 37 (28%) of 158 PWE died over a 10-year period [25]. In the latter study, the overall relative risk of dying in PWE over a 10-year period compared to controls was 6.2 (95% CI: 2.7–14.1) [25]. In our study, five OIPWE died because their parents refused to take them to the clinic and preferred traditional treatment instead of AED. To avoid such deaths, the inclusion of traditional healers in epilepsy treatment programs should be considered [26]. 

In our study, several adverse events were observed, most of which were minor and transitory. The frequency of drug-related adverse events was similar across the different treatment arms. Nonetheless, one OIPWE developed TEN induced by phenobarbital. TEN has previously been observed with several AED including phenobarbital [27,28,29]. Our study illustrates the importance of counseling PWE, caretakers, CHW and local health personnel about early identification of adverse events caused by AED to ensure prompt discontinuation of potentially harmful drugs. One mother who did not receive ivermectin during pregnancy but who was treated with phenobarbital gave birth to a child with a cleft lip. Phenobarbital, as all first line AED available in sub-Saharan Africa, has been associated with congenital abnormalities [30]. Therefore, more advocacy is needed to make safer second line AED [31,32] available at an affordable price. 

This study is not void of limitations. The fact that PWE were not blinded to the number of ivermectin treatments they received could have influenced the observed outcomes. The required sample size to detect the hypothesized difference in probability of seizure freedom among individuals with administration of ivermectin twice or thrice was not reached. Differences in inclusion criteria, regarding the number of seizures at baseline, for individuals in the two groups as well as differences in ivermectin treatment regimens within each of the groups complicate the interpretation of the results in a combined analysis in which the treatment effect in terms of the doses administered is apparently considered similar across the groups. However, due to the different treatment regimens in the two groups, the combined model is fully saturated with regard to the group and treatment regimen interaction. Consequently, the model does not impose a common treatment effect in the two groups.

Moreover, up to nine participants were switched to another AED, and in 65 others, the AED dose was increased >30 mg during the trial because seizures were not controlled. An increase of the AED dose >30 mg was associated with a more severe form of epilepsy (as AED increase was more frequent in PWE from group 1 than in group 2) and of seizures uncontrolled by ivermectin. Indeed, increase in AED dose was associated with a lower probability of achieving seizure freedom, possibly because the neurological damage in these PWE was severe, rendering the seizures difficult to manage even with high AED doses. Finally, PWE who became pregnant during the study were not given further doses of ivermectin because according to international guidelines ivermectin should not be given to pregnant women [12]. 

In conclusion, this study, in the combined analysis of group 1 and 2 participants, shows the advantage of treating OIPWE with ivermectin at least bi-annually in addition to regular AED. Moreover, it lends support to the accumulating evidence that infection with *O. volvulus* can trigger seizures. Increasing the frequency of community-directed treatments with ivermectin in areas with ongoing onchocerciasis transmission and high epilepsy prevalence will not only decrease the incidence of OAE, but also improve the quality of life of OIPWE while accelerating the elimination of onchocerciasis. 

## Figures and Tables

**Figure 1 pathogens-09-00205-f001:**
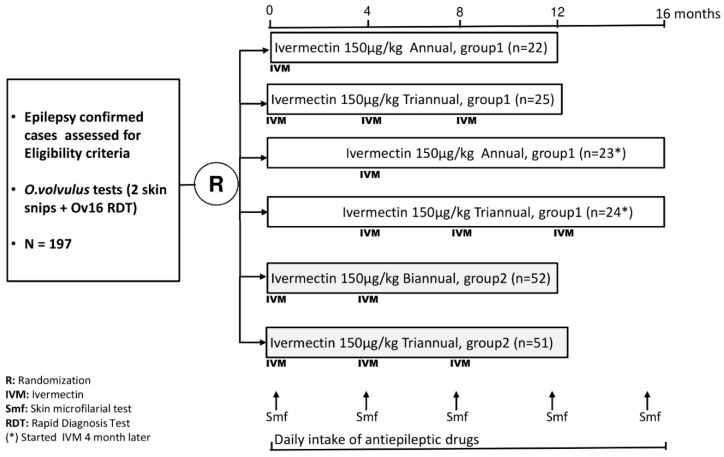
Study participants’ recruitment and follow-up plan.

**Figure 2 pathogens-09-00205-f002:**
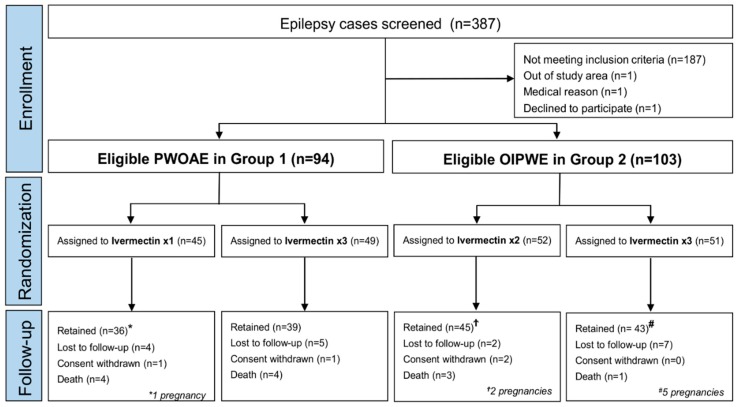
Trial profile of screened, randomized, treated, and analyzed patients by study group.

**Figure 3 pathogens-09-00205-f003:**
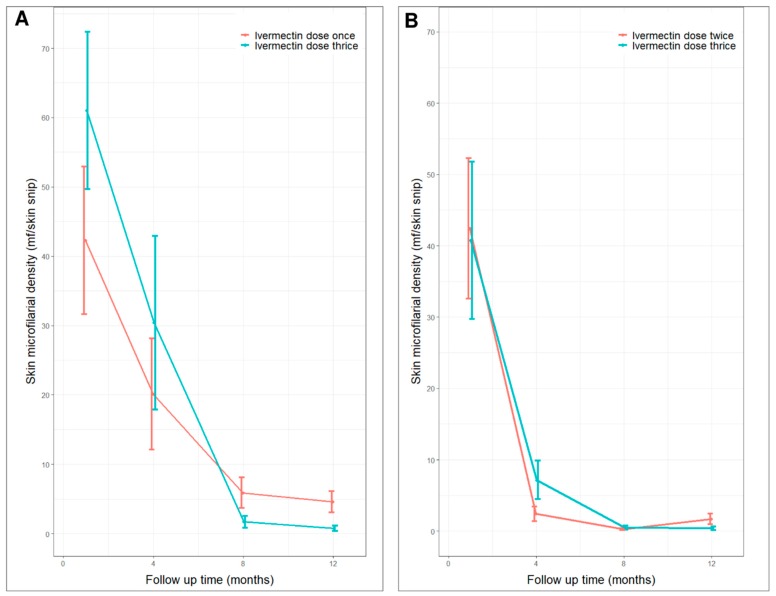
Average skin microfilarial density (with standard error bars) at baseline, month 4, 8 and 12. (**A**) Group 1 participants (ivermectin thrice vs ivermectin once); (**B**) Group 2 participants (ivermectin thrice vs ivermectin twice).

**Figure 4 pathogens-09-00205-f004:**
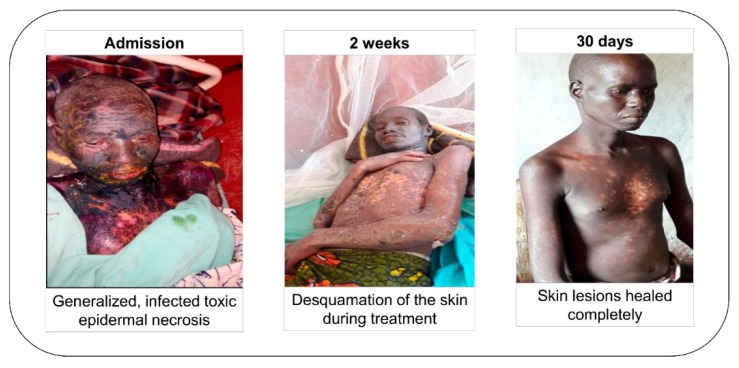
Evolution of the toxic epidermal necrosis, from hospital admission up to discharge.

**Table 1 pathogens-09-00205-t001:** Baseline characteristics of the participants in the different treatment arms.

	Group 1			Group 2			*p*-Value *
	Ivermectin x1 (n = 45)	Ivermectin x3 (n = 49)	Total (n = 94)	Ivermectin x2 (n = 52)	Ivermectin x3 (n = 51)	Total (n = 103)	
Median age in years (IQR)	24 (18–30)	22 (16–28)	20 (16–27)	24 (17–35)	28 (21–35)	26 (18–35)	0.001
Males, n (%)	29 (64)	29 (59)	58 (62)	21 (40)	22 (43)	43 (42)	0.006
Height in cm, median (IQR)	150 (144–159)	152 (146–158)	152 (144–158)	155 (150–163)	154 (146–160)	155 (147–162)	0.072
Weight in kg, median IQR)	42 (34–51)	47 (38–50)	45 (35–51)	49 (44–54)	45 (40–51)	48 (41–53)	0.112
Ivermectin use in the past, n (%)	0 (0)	0 (0)	0	2 (4)	1 (2)	3 (3)	0.247
***Seizure characteristics:***							
History of generalized motor seizures, n (%) **	43 (95)	46 (95)	89 (95)	52 (100)	49 (96)	101 (98)	0.261
History of absence seizures, n (%)	26 (58)	29 (47)	55 (59)	14 (27)	18 (35)	32 (31)	0.002
History of nodding seizures, n (%)	5 (11)	2 (3)	7 (8)	1 (2)	1 (2)	2 (2)	0.089
Seizure frequency per month, median (IQR)	2 (1–4)	3 (2–7)	3 (2–6)	1 (1–4)	2 (1–3)	2 (1–3)	<0.001
Epilepsy duration in years, median (IQR)	10 (6–15)	13 (7–17)	12 (16–27)	6 (3–14)	10 (2–17)	7 (3–16)	0.093
***Physical examination:***							
Altered general state, n (%)	23 (51)	18 (30)	41 (44)	6 (11)	12 (24)	18 (17)	0.001
Itching, n (%)	22 (49)	19 (40)	41 (44)	18 (35)	21 (41)	39 (38)	0.468
Palpable nodules, n (%)	3 (6)	7 (10)	10 (11)	5 (10)	3 (6)	8 (8)	0.621
Burn scars, n (%)	17 (38)	19 (30)	36 (38)	10 (19)	11 (22)	21 (20)	0.007
Leopard skin, n (%)	1 (2)	1 (2)	1 (1)	2 (4)	1 (2)	3 (3)	0.734
Lizard skin, n (%)	4 (8)	4 (7)	8 (9)	1 (7)	1 (2)	2 (2)	0.234
Cognitive impairment, n (%)	21 (47)	22 (33)	43 (46)	10 (19)	11 (22)	21 (20)	0.002
Microfilarial density per skin snip, median (IQR)	11 (0–53)	28 (3–48)	17 (1–72)	6 (0–78)	6 (1–37)	6 (0–56)	0.062
Skin snip positivity, n (%)	33 (73)	40 (79)	73 (78)	39 (56)	29 (57)	68 (66)	0.078
Ov16 positivity	32 (71)	35 (71)	67 (71)	42 (81)	35 (63)	77 (75)	0.338

Group 1: OIPWE meeting the OAE criteria with ≥2 seizures/month at baseline, Group 2: OIPWE not meeting the OAE criteria or <2 seizures/month at baseline; * Fisher exact test for counts and median test for continuous variables comparing baseline characteristics of group 1 (Total) and 2 (Total) participants; IQR: Interquartile range. ** Includes tonic-clonic and atonic generalized seizures.

**Table 2 pathogens-09-00205-t002:** Characteristics of the participants at month 12 of follow-up in the different treatment arms.

	Group 1			Group 2			*p*-Value *
	Ivermectin x1	Ivermectin x3	Total	Ivermectin x2	Ivermectin x3	Total	
***Status of participants at 12 months:***							
Intent-to-treat set, n (%)	45 (23)	49 (51)	94 (100)	52 (26)	51 (26)	103 (100)	
As-treated set, n (%)	41 (91)	39 (79.6)	80 (85)	49 (94.2)	37 (72.5)	86 (83.5)	
Attended last follow-up visit, n (%)	36 (91)	39 (79.6)	75 (79.7)	45 (86.5)	43 (84.3)	88 (85.4)	0.139
Lost to follow up, n (%)	4 (8.8)	5 (10.2)	9 (9.6)	2 (3.8)	7 (13.7)	9 (8.7)	0.838
Consent withdrew, n (%)	1 (2.2)	1 (2)	2 (4.1)	2 (3.8)	0 (0)	2 (2.0)	0.117
Death, n (%)	4 (8.9)	4 (8.2)	8 (8.5)	3 (5.8)	1 (1.9)	4 (3.9)	0.235
Pregnant, n (%)	1 (2)	0 (0)	1 (1)	2 (3.8)	5 (10)	7 (6.8)	0.035
Seizure-free during last four months, n (%)	12 (27)	21 (43)	33 (35)	31 (59)	21 (41)	52 (50)	0.034
Seizure frequency during last four months, median (IQR)	2 (0–5)	1 (0–3)	2 (0–5)	0 (0–2)	0 (0–2)	0 (0–2)	0.016
>50% seizure reduction during last four months, n (%)	18 (40)	27 (55.1)	45 (47.9)	34 (65.4)	28 (54.9)	62 (60.2)	0.227
Optimal adherence to AED during last four months, n (%)	28/45 (62)	29/49 (59)	57 (61)	31/51 (57)	36/52 (77)	67 (65)	0.562
Increased AED dose >30 mg between M0 and M12	21/45 (47)	18/49 (37)	39/94 (41)	14/52 (27)	12/51 (23)	26 (24)	0.022
Switch to a different AED, n (%)	2/45 (4)	4/49 (8)	6 (6)	0/52 (0)	3/51 (6)	3 (3)	0.308
***Physical examination:***							
Altered general state, n (%)	5 (14)	7 (14)	12 (17)	5 (10)	8 (17)	13 (14)	0.582
Itching, n (%)	3 (8)	5 (5)	8 (9)	0 (0)	0 (0)	0	0.003
Burn scars, n (%)	1 (3)	2 (5)	3 (3)	0 (0)	2 (4)	2 (2)	0.669
Skin snip positivity, n (%)	16 (44)	7 (15)	23 (31)	13 (32)	5 (11)	18 (19)	0.065
Microfilarial density per skin snip, median (IQR)	0 (0–3)	0 (0–0)	0 (0–2)	0 (0–1)	0 (0–0)	0 (0–1)	0.126

Group 1: OIPWE meeting the OAE criteria with ≥2 seizures/month at baseline, Group 2: OIPWE not meeting the OAE criteria or <2 seizures/month at baseline; * Fisher exact test for counts and median test for continuous variables comparing the characteristics of group 1 (Total) and 2 (Total) participants at month 12 of follow-up; IQR: Interquartile range; AED: Anti-epileptic drug.

**Table 3 pathogens-09-00205-t003:** Multiple logistic regression exploring the association between ivermectin dose thrice vs once and seizure freedom during M9 to M12 of follow-up (group 1).

Variables	OR	95% CI		*p*-Value
Ivermectin dose: thrice vs once	2.310	0.865	6.592	0.111
Female vs male	1.418	0.521	3.939	0.505
Age	1.070	0.981	1.175	0.156
Weight (in kg)	1.020	0.968	1.079	0.476
Seizure frequency at baseline	0.667	0.301	1.341	0.291
Duration of epilepsy (years)	0.604	0.263	1.355	0.236
Microfilarial density at baseline *	0.779	0.591	1.009	0.073
Optimal vs sub-optimal AED adherence during last four months	3.358	1.206	10.181	0.029
>30 mg of AED increase (between M0 and M12) vs no increase	0.278	0.087	0.797	0.026

OR: Odds ratio; CI: Confidence interval; AED: Antiepileptic drug; *: Log-transformed.

**Table 4 pathogens-09-00205-t004:** Multiple logistic regression exploring the association between ivermectin thrice and twice and seizure freedom during M9 to M12 of follow-up (group 2).

Variables	OR	95% CI		*p*-Value
Ivermectin dose thrice vs twice	0.477	0.196	1.125	0.107
Female vs male	0.871	0.347	2.131	0.772
Age	0.988	0.946	1.030	0.583
Weight (in kg)	0.990	0.941	1.041	0.713
Seizure frequency at baseline	0.823	0.458	1.462	0.527
Duration of epilepsy (years)	1.105	0.688	1.812	0.693
Microfilarial density at baseline *	0.739	0.573	0.934	0.018
Optimal vs sub-optimal AED adherence during last four months	2.792	1.141	7.281	0.034
>30 mg of AED increase (between M0 and M12) vs no increase	0.212	0.067	0.595	0.006

OR: Odds ratio; CI: Confidence interval; AED: Antiepileptic drug; *: Log-transformed.

**Table 5 pathogens-09-00205-t005:** Multiple logistic regression exploring the association between taking ivermectin thrice, twice, or once, and seizure freedom during M9 to M12 of follow-up adjusted for study group membership (combined analysis for group 1 and group 2).

Variables	OR	95% CI		*p*-Value
Ivermectin dose twice vs once	5.087	1.378	19.749	0.018
Ivermectin dose thrice vs once	2.471	0.944	6.769	0.075
Female vs male	1.152	0.593	2.235	0.681
Age	0.999	0.964	1.036	0.967
Weight (in kg)	1.004	0.969	1.039	0.838
Seizure frequency at baseline	0.731	0.460	1.133	0.180
Duration of epilepsy (years)	1.039	0.711	1.538	0.848
Microfilarial density at baseline *	0.752	0.627	0.892	0.002
Optimal vs sub-optimal AED adherence during last four months	3.274	1.673	6.635	0.001
>30 mg of AED increase (between M0 and M12) vs no increase	0.253	0.116	0.525	<0.001
Study group 1 vs group 2	1.989	0.749	5.466	0.182

OR: Odds ratio; CI: Confidence interval; AED: Antiepileptic drug; *: Log-transformed.

**Table 6 pathogens-09-00205-t006:** Multiple logistic regression exploring the association between taking ivermectin thrice, twice, or once, and >50% seizure reduction during M9 to M12 of follow-up (combined analysis for group 1 and group 2).

Variables	OR	95% CI		*p*-Value
Ivermectin dose twice vs once	4.469	1.250	16.620	0.026
Ivermectin dose thrice vs once	2.693	1.077	6.998	0.042
Female vs male	1.049	0.544	2.024	0.890
Age	1.006	0.970	1.045	0.759
Weight (in kg)	0.991	0.956	1.027	0.631
Seizure frequency at baseline	1.445	0.915	2.355	0.134
Duration of epilepsy (years)	1.087	0.744	1.590	0.675
Microfilarial density at baseline *	0.857	0.720	1.016	0.087
Optimal vs sub-optimal AED adherence during last four months	1.431	0.734	2.799	0.305
>30 mg of AED increase (between M0 and M12) vs no increase	0.250	0.122	0.496	<0.001
Study group 1 vs group 2	1.634	0.609	4.454	0.344

OR: Odds ratio; CI: Confidence interval; AED: Antiepileptic drug; *: Log-transformed.

**Table 7 pathogens-09-00205-t007:** Multivariable analysis investigating the effect of absence of mf during M9 to M12 of follow-up on seizure freedom during the same period.

Variables	OR	95% CI		*p*-Value
Absence vs presence of mf during M9 and M12 follow-up visits	2.618	1.136	6.289	0.027
Female vs male	1.089	0.539	2.187	0.811
Age (years)	0.992	0.949	1.038	0.711
Weight (in kg)	1.013	0.977	1.051	0.477
Seizure frequency at baseline	0.719	0.432	1.175	0.192
Duration of epilepsy (years)	1.001	0.952	1.053	0.960
Optimal vs sub-optimal AED adherence during last four months	1.885	0.909	3.988	0.092
>30 mg of AED increase (between M0 and M12) vs no increase	0.193	0.088	0.407	<0.001
Study group 1 vs group 2	0.861	0.391	1.912	0.711

OR: Odds ratio; CI: Confidence interval; AED: Antiepileptic drug; Mf: microfilariae.

**Table 8 pathogens-09-00205-t008:** Multiple logistic regression exploring the association between ivermectin thrice and once and seizure freedom during M9 to M12 of follow-up (as-treated analysis, group 1).

Variables	OR	95% CI		*p*-Value
Ivermectin dose thrice vs once	3.318	1.161	10.500	0.035
Female vs male	1.526	0.521	4.671	0.458
Age	1.061	0.967	1.177	0.248
Weight (in kg)	1.011	0.952	1.074	0.732
Seizure frequency at baseline	0.815	0.347	1.755	0.627
Duration of epilepsy (years)	0.681	0.291	1.553	0.380
Microfilarial density at baseline *	0.746	0.542	0.999	0.065
Optimal vs sub-optimal AED adherence during last four months	1.897	0.635	6.001	0.278
>30 mg of AED increase (between M0 and M12) vs no increase	0.201	0.063	0.581	0.006

OR: Odds ratio; CI: Confidence interval; AED: Antiepileptic drug; *: Log-transformed.

**Table 9 pathogens-09-00205-t009:** Multiple logistic regression exploring the association between ivermectin thrice and twice and seizure freedom during M9 to M12 of follow-up (as-treated analysis, group 2).

Variables	OR	95% CI		*p*-Value
Ivermectin dose thrice vs twice	0.484	0.168	1.348	0.183
Female vs male	0.851	0.306	2.279	0.758
Age	0.988	0.943	1.034	0.630
Weight (in kg)	0.980	0.924	1.037	0.508
Seizure frequency at baseline	0.775	0.388	1.521	0.480
Duration of epilepsy (years)	1.009	0.592	1.756	0.975
Microfilarial density at baseline *	0.657	0.484	0.859	0.005
Optimal vs sub-optimal AED adherence during last four months	2.107	0.767	6.075	0.170
>30 mg of AED increase (between M0 and M12) vs no increase	0.147	0.041	0.451	0.002

OR: Odds ratio; CI: Confidence interval; AED: Antiepileptic drug; *: Log-transformed.

**Table 10 pathogens-09-00205-t010:** Multiple logistic regression exploring the association between ivermectin once, twice, and thrice and seizure freedom during M9 to M12 of follow-up adjusted for study group (as-treated analysis, group 1 and 2 combined).

Variables	OR	95% CI		*p*-Value
Ivermectin dose twice vs once	10.033	2.670	42.496	0.001
Ivermectin dose thrice vs once	4.795	1.790	14.089	0.003
Female vs male	1.135	0.548	2.348	0.738
Age (years)	0.999	0.960	1.038	0.947
Weight (in kg)	0.993	0.954	1.033	0.731
Seizure frequency at baseline	0.771	0.453	1.279	0.333
Duration of epilepsy (years)	1.031	0.686	1.574	0.886
Microfilarial density at baseline *	0.691	0.561	0.837	0.000
Optimal vs sub-optimal AED adherence during last four months	1.966	0.935	4.238	0.085
>30 mg of AED increase (between M0 and M12) vs no increase	0.171	0.074	0.372	<0.001
Study group 1 vs group 2	3.141	1.097	9.668	0.043

OR: Odds ratio; CI: Confidence interval; AED: Antiepileptic drug; *: Log-transformed.

**Table 11 pathogens-09-00205-t011:** Adverse events reported during the 12 months of follow-up.

	Ivermectin x 3	Ivermectin x 2	Ivermectin x 1	0verall
Number of participants	100	52	45	197
Ae reported	84	44	41	169
Average number of AE per person	0.84	0.85	0.91	0.86
AE possibly related to the treatment, n (%)	64 (76)	37 (84)	32 (78)	133 (79)
**AE severity**				
Minimal, n (%)	31 (37)	14 (32)	11 (27)	56 (33)
Moderate, n (%)	34 (41)	21 (48)	20 (49)	75 (44)
Severe, n (%)	17 (20)	6 (14)	10 (24)	33 (20)
Life-threatening, n (%)	1 (1)	2 (5)	0	3 (2)
Fatal outcome, n (%)	5 (6)	3 (7)	4 (10)	12 (7)
Any SAE, n (%)	13 (16)	3 (7)	4 (10)	20 (12)
SAE possibly related to the treatment, n (%)	9 (11)	1 (2)	2 (5)	12 (7)

AE: adverse event; SAE: serious adverse events.

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
