# Peer review of "Single versus Multiple Dose Ivermectin Regimen in Onchocerciasis-Infected Persons with Epilepsy Treated with Phenobarbital: A Randomized Clinical Trial in the Democratic Republic of Congo"

_pathogens, 2020, doi:10.3390/pathogens9030205_

Round 1

Reviewer 1 Report

The paper by Mandro et al reports on the effects of ivermectin (IVM) on the incidence of seizures in onchocerciasis-infected persons with epilepsy (OIPWE) in rural Democratic Republic of the Congo. The authors found that OIPWE who receive IVM trice per year have higher odds of seizure freedom than those treated once or thrice a year. The antiseizure effect of IVM was associated with microfilarial density. Thus, they conclude that these findings suggest that O.volvulus infection play an etiological role in causing seizures.

Although the findings are promising, the anticonvulsant effect of IVM has been reported (Kipp et al., Lancet 340:789-90. The authors should emphasize on the facts that in their study IVM suppress absence seizures and nodding seizures. Then in the Discussion, they should then speculate on possible mechanisms involving thalamo-cortical networks for absence seizures and limbic circuits for nodding seizures.

Introduction

The authors should provide more information about publish papers related to their work. For instance, a relevant publication by co-author Mukendi et al, 2019, Int J Infect Dis 79:187-194 is not included or discussed.

Results

Table 1.

History of generalized seizures. What type? Generalized tonic-clonic seizures? Generalized clonic seizures?

History of absence seizures. Is this "pure" absence seizure or absence seizures with motor components?

Table 2

Seizure-free during the last four months.  The seizures are absence seizures, nodding seizures or both.

Table 5

Duration of epilepsy. Please clarify the epilepsy type. Absence epilepsy? Nodding epilepsy or both or other types of epilepsy.

Discussion.

The authors should discuss potential mechanisms of IVM underlying the anticonvulsant effect against absence seizures. Multiple lines of evidence indicate that IVM potentiates inhibition mediated by glutamate-gated chloride channel receptors (GluClRs) activity (PLOs Pathol 15, e1007570. Other mechanisms may include pH regulation, modulation of nicotinic receptors, activation of GABA-A receptors, activation of G-protein-gated inwardly K+ (GIRK) channels etc...

There is evidence that IVM causes CNS depression.

Interestingly, there are experimental evidence that IVM suppressed lidocaine- and strychnine-induced seizures (Vet res Commun 31:863, 2007). This paper should be included and discussed.

Author Response

Response to REVIEWER #1

Point 1. The paper by Mandro et al reports on the effects of ivermectin (IVM) on the incidence of seizures in onchocerciasis-infected persons with epilepsy (OIPWE) in rural Democratic Republic of the Congo. The authors found that OIPWE who receive IVM trice per year have higher odds of seizure freedom than those treated once or thrice a year. The antiseizure effect of IVM was associated with microfilarial density. Thus, they conclude that these findings suggest that O. volvulus infection play an etiological role in causing seizures.

Although the findings are promising, the anticonvulsant effect of IVM has been reported (Kipp et al., Lancet 340:789-90. The authors should emphasize on the facts that in their study IVM suppress absence seizures and nodding seizures. Then in the Discussion, they should then speculate on possible mechanisms involving thalamo-cortical networks for absence seizures and limbic circuits for nodding seizures.

Response: Our findings suggest that IVM supresses mainly generalised tonic-clonic seizures (see new supplementary Table below, has been added to the manuscript as S4). We do not speculate about possible mechanisms because it is unlikely that ivermectin itself penetrates the CNS, as it cannot cross the human blood-brain barrier (Ref: Serious Neurological Adverse Events after Ivermectin-Do They Occur beyond the Indication of Onchocerciasis? Am J Trop Med Hyg 2018, 98, 382-388, doi:10.4269/ajtmh.17-0042). Therefore, it seems more plausible to orientate the discussion towards the fact that IVM decreases seizure frequency by clearing microfilariae, as previously suggested by Siewe Fodjo et al (Ref: Onchocerciasis-associated epilepsy in the Democratic Republic of Congo: Clinical description and relationship with microfilarial density. PLoS Negl Trop Dis. 2019; 13: e0007300. doi:10.1371/journal.pntd.0007300).

S4 Table : Seizure characteristics of participants at baseline and during the last four months of the study

Enrolled at baseline

N = 197

Completed last four months’ evaluation

N = 157

Premature discontinuation of the study*

N = 40

Seizure Types

Baseline Seizures:

n (% participants)

Seizure-free during the last four months:

n (% baseline)

Deaths

Lost to follow-up

Withdrew consent

Tonic-clonic only

105 (53.3%)

48 (45.7%)

6

9

5

Atonic only

1 (0.5%)

0 (0%)

1

0

0

Absence seizures only

4 (2%)

2 (50%)

0

2

0

Nodding seizures only

1 (0.5%)

0 (0%)

0

0

0

Absences+Tonic-clonic seizures

70 (35.5%)

31 (44.3)

3

5

4

Nodding+Tonic-clonic seizures

4 (2%)

1 (25%)

0

1

0

Nodding+Absence seizures

4 (2%)

1 (25%)

1

0

0

Other seizures types (Focal, etc)

8 (4.1%)

2 (25%)

1

2

0

Total

197

85

12

19

9

*Considered as not having achieved seizure freedom (treatment failure)

Point 2.

Introduction

The authors should provide more information about publish papers related to their work. For instance, a relevant publication by co-author Mukendi et al, 2019, Int J Infect Dis 79:187-194 is not included or discussed.

Response : We now cite this paper in the introduction of our manuscript.

Point 3.

Results

Table 1.

History of generalized seizures. What type? Generalized tonic-clonic seizures? Generalized clonic seizures?

Response: The generalized seizures encountered during our study were: Tonic-clonic and atonic seizures. This has now been specified in Table 1.

Point 4.

History of absence seizures. Is this "pure" absence seizure or absence seizures with motor components?

Response: These are not pure absences, as most of these participants also had other seizure types associated. We have now added a supplementary Table which details the number of PWE with pure absences (See Table S4 above).

Point 5.

Table 2

Seizure-free during the last four months.  The seizures are absence seizures, nodding seizures or both.

Response: Seizure-free means the participant does not experience any seizure type, be it generalized, nodding, or absences. We now specify this.

Point 6.

Table 5

Duration of epilepsy. Please clarify the epilepsy type. Absence epilepsy? Nodding epilepsy or both or other types of epilepsy.

Response: Duration of epilepsy refers to any seizure type, be it generalized, nodding, or absences.

Point 7.

Discussion.

The authors should discuss potential mechanisms of IVM underlying the anticonvulsant effect against absence seizures. Multiple lines of evidence indicate that IVM potentiates inhibition mediated by glutamate-gated chloride channel receptors (GluClRs) activity (PLOs Pathol 15, e1007570.

Other mechanisms may include pH regulation, modulation of nicotinic receptors, activation of GABA-A receptors, activation of G-protein-gated inwardly K+ (GIRK) channels etc...

There is evidence that IVM causes CNS depression.

Interestingly, there are experimental evidence that IVM suppressed lidocaine- and strychnine-induced seizures (Vet res Commun 31:863, 2007). This paper should be included and discussed.

Response:

We now cite the papers suggested by the reviewer, and include the following paragraph in the discussion:

“Ivermectin interacts with the glutamate-gated chloride channel receptors (GluClRs) activity that mediate neuronal and muscular inhibition [15]. However GluCIRs are expressed exclusively in invertebrates [15]. In the mammalian central nervous system (CNS), ivermectin interacts with at least three targets: a GABA-dependent chloride channel, a glycine-dependent chloride channel and a voltage-dependent channel [16,17]. Through these interactions ivermectin may cause a dose-depended CNS depression (including seizure suppression) in rats [17,18]. However, the penetration of ivermectin into murine CNS must be interpreted in the light of evidence demonstrating an increased permeability of their blood-brain barrier (BBB) during the neonatal period [19]. Ivermectin given at therapeutic doses is unlikely to cross the human BBB [20] and therefore cannot elicit a direct anti-epileptic effect in the CNS.”

Reviewer 2 Report

Thank you very much for your submission and work in the area. Much needed. Would you be able to use the seizure reduction data you collected into a seizure frequency score?

Author Response

Response to REVIEWER #2

Point 1

Thank you very much for your submission and work in the area. Much needed. Would you be able to use the seizure reduction data you collected into a seizure frequency score?

Response:

Given that we did not collect all the necessary data to formulate epilepsy severity scores, we prefer to stick to the primary endpoint as was mentioned in the analysis plan in the protocol of the trial (https://clinicaltrials.gov/ct2/show/NCT03852303). Nonetheless, seizure frequency (which we used as outcome parameter) is already a major component of most seizure severity scores (Ref: Quantitative assessment of seizure severity for clinical trials: a review of approaches to seizure components. Epilepsia. 2001 Jan;42(1):119-29. DOI: 10.1046/j.1528-1157.2001.19400.x), and could be used as a proxy for seizure severity.

Reviewer 3 Report

1. This is a phase-4 study with two groups (1 & 2), each with two different medication regimens (Group 1, once versus thrice;  Group 2, twice versus thrice).

2. This study started as a pilot trial of the effect of ivermectin administered once versus thrice “per” year on seizure freedom during the last four months of the 12 months trial.

>I suggest avoiding the term “per year”, since “per” inadvertently suggests a study duration lasting several years, and this wording also implicitly suggests that continued medication beyond 12 months has an effect, whereas this was not investigated.

3. This pilot trial was registered at clinicaltrials.gov with number NCT03852303 (in this review referred to as ‘submitted protocol’); the registry indicates the trial’s status as: “completed (Actual Primary Completion Date: March, 2019; Actual Study Completion Date: July 1, 2019), with time frame for primary endpoint month 9-12, originally estimated enrollment n=150; patients included n=94; participants actually enrolled n=197; results not reported”.

4. The primary endpoint of the study was seizure freedom during the last four months.

>This can only be a valid end point if groups for comparison each had a similar pre-trial seizure frequency.

>Having seizure freedom during the last four months of the trial as an endpoint, requires a comparison with a sequential series of pre-trial four months periods, for example over one pre-trial year, during which at least the number of seizure-freedom periods of four months had been monitored.

>Since Group 1 and Group 2 had different inclusion criteria with respect to pre-trial seizure frequency per months (less than 2 or even zero?), such comparison per individual requires not only seizure freedom per 4 months, but also seizure frequency!

>In anti-epileptic drug trials, a commonly used definition of clinical effect is 50% reduction of seizures per pre-defined period in at least 50% of patients. How are results when the data are analyzed in this way, taking also into account the previous remarks under 4.? See also remarks at 5. (below)

5. The initial set of inclusion criteria included a pre-trial seizure frequency of two or more seizures per month. (See also point 8, below)

>This inclusion criterion is mentioned in the manuscript, but NOT in the submitted protocol;

>This inclusion criterion is a relevant component of study design since reduction of seizures to none during the last four months of the study (month 9-12 according to submitted protocol) is the primary endpoint, and the pre-trial seizure frequency affects statistical power;

>The length of observation of pre-trial seizures needs to be specified;

>Discarding the inclusion criterion of two or more pre-trial seizures per month for inclusion into Group 2 obviously would result in reduction of power to observe an effect of the trial medication.

>The difference in inclusion criteria between Group 1 and Group 2 – seizure frequency - directly relates to the endpoint (seizure freedom) and may have introduced significant bias. How controlled for this?

>See also remarks under 4. (above)

6. Discrepancies in numbers between submitted protocol, figures 1 and 2, and text: Group 1: Figure 1 counts n=44 exposed once (21+23) and 46 exposed thrice (22+24), total of n=90, whereas the text (P5/Lines 196-200) counts 45 exposed once and 49 exposed thrice, total of n=94. Group 2: Figure 1 counts n=30 exposed twice and n=30 exposed thrice, whereas the text (P5/Lines 196-200) counts n=52 exposed twice and n=51 exposed thrice total of n=103.

>This needs explanation or correction, and check for impact on analysis results, if any.

>It seems that the data Figure 1 represents not a flow diagram of the study as conducted, but reflects more of the originally estimated inclusion, and not the actual inclusion after ad interim adjustment of the protocol.

>Further confusion arises by the text on P2/Line60, where the intended inclusion is given as n=110, a number that cannot be derived by any figure in the submitted protocol, figures or text elsewhere in this manuscript.

>In the submitted protocol, the originally estimated inclusion is given as n=150. Figure 2: re 100 participants treated thrice: “24 did not receive all three doses” (P7/L220) “10 missed a dose because not visiting the health center” (P7/221)

>>Were the 10 missing a dose included in the 24 not receiving all three doses?

>>Were these 24 and 10 excluded from further analysis? If not, or differentially, how was skipping one or two doses taken into account in the analysis?

>>What was the reason for not receiving all three doses in the 24? (Why is this specified for the 10 missing one dose, and not for the 24?)

>Thrice treated group: N=100 (49 from Group 1 + 51 from Group 2) The text specifies lost to follow-up (n=12), consent withdrawn (n=6), death n=5).

>>Assuming that these three categories are mutually exclusive and subtracting the total of these three (n=23) from n=100 participants, the number of retained participants would be n=77 and not n=80.

7. The first group (Group 1) started as pilot study in line with the submitted protocol, but continued with randomization towards once versus thrice. The second group (Group2) comparing twice versus thrice “per” annuum was added afterwards. This was done in order to cope with the problem of low recruitment, apparent lack of statistical power, and perceived pressure from the community i.e. the population of onchocerca infected persons with epilepsy (OIPWE), to participate, likely because of benefits expected from these subsequent candidate participants.

>This may well have resulted in a more optimistic evaluation of outcome among participants in Group 2 compared to Group 1.

8. It is good to note that the study design is one that compares different medication regimens, thus WITHOUT a placebo control.

>In the discussion section, the authors state that the negative results of a previous study of effect of ivermectin on seizure frequency can be explained by the lack of a placebo control in that study.

>In fact, also the current study lacks placebo controls, to be stated as such in this manuscript and not to be contrasted with a previously published study with negative outcome;

>Furthermore, participants were aware of differences in medication regimens (once versus twice versus thrice);

>In this respect, Group 2 is incomparable to Group 1 because of the pressure from that population to ‘benefit’ from participation, and the loosening of the inclusion criteria.

>Both may have induced a significant placebo effect.

9. There is no mention of a possible role of pharmacokinetic interaction between ivermectin and phenobarbital (literature!).

>These interactions may be quite complex and perturb the interpretation of results.

>This is especially relevant since seizure frequency is the primary outcome measure and may be influenced also by phenobarbital levels.

>Effect of ivermectin on seizure frequency by effect on phenobarbital levels (induction of metabolism) can only be excluded if this has been sorted out in previous studies, or by plasma level monitoring of phenobarbital during the current study.

>If none is available, that would need caveat remarks in the discussion and conclusion.

>Vice versa, phenobarbital may have a negative effect on e.g. gut mucosa PGP/MDR1 mediated bioavailability of ivermectin and accelerate the catabolism of ivermectin by induction of CYP3A. How did the investigating team take these possible interactions into account while setting up the study and interpreting the data as obtained?

10. The inclusion criteria related to seizures/epilepsy is markedly different between Group 1 and Group 2. (see also point 4). Apparently for both group 1 and 2, the ILAE definition is used, of at least two seizure separated more than 24 hours (Quota from Page 3, lines 85-88: “To be enrolled, participants had to meet the 2014 International League Against Epilepsy (ILAE) definition of epilepsy: having experienced at least two seizures, unprovoked and without fever, with a minimal time difference of 24 hours between the two events”), thus without setting a minimum per month.

>Only for the initial pilot study (n=94), from which group 1 evolved, there was an additional inclusion criterion of at least two seizures per month. This makes Group 1 and Group 2 incomparable with respect to seizure outcome measures and statistical power calculation. For both group 1 and 2, more data is needed concerning length of pre-trial observation of seizures and calculation of seizure frequency.

11. The inclusion criteria are not consistent.

>See legend to figure 2: “OIPWE not meeting the OAE criteria“or” < 2 seizures/month at baseline”. Thus, according to figure 2, subjects were included that not had onchocerca-associated epilepsy. Why then, were they included? Or change the wording?

12. Skin snip positivity and/or [antibody] Ov16 positivity (see methods as well as table 1).

>This implies that patients were regarded as having active onchocerca, whereas only antibodies were positive. Is this correct or, what is the justification for this? Please specify numbers for each of the four combination of the two (Skin snip pos/neg & ab-OV16 pos/neg)(+/+, +/-, -/+, -/-).

13. As expected, group 2 patients had less seizures and in general less disease related features at inclusion.

>In this respect, also the difference in pregnancy rate is not surprising.

14. In Table 2 evaluation data are given at month 12. According to the scheme in Figure 1, the thrice medicated group consisted of three subgroup For one of the three subgroups (Group 1, pilot trial), this evaluation was done three months after the third dosing of ivermectin, whereas for two of the three subgroups (Group 1, after re-randomization; Group 2, one arm) this was done three months after the second dosing, at or around the time of the third dosing.

>If so, please specify whether the evaluation in the latter was shortly before or how long after the third dose.

>If not, please specify whether evaluation at 12 months is related to point of time of entry into the study OR the start of medication.

>The last latter be the only appropriate comparison (equal duration of treatment in relation to time of evaluation).

>For clarity, throughout the manuscript, it is better to use consistently Preset points in time (e.g. T=0, T=4, T=8, etc) than to mix points in times with durations. In this way, durations (of exposure) can be derived uniformly.

15. Was the skin snip done always shortly before each dosing of ivermectin (appropriate), afterwards, or irregularly, or otherwise?

16. Figure 3. At baseline and at 8 and 12 months of the once per year treatment arm, Group 1 (A) was not different from that of the twice and thrice arm, Group 2 (B).

>This leaves the question unanswered whether there is a significant difference at 4 months and, if so, how to interpret its clinical significance.

17. P4/L109-110: “The minimal time interval between two consecutive doses of ivermectin was four months.”

>What was, in practice during this study, the variation in time interval?

>What was the maximum interval between two successive doses allowed for continuation of study participation?

>Was there a difference between study-groups and -arms that may have had an effect on interpretation of the results?

18. P4/L112-116: “The AED (phenobarbital) dose was based on the participants’ weight: 5mg/kg for participants weighing <15kg; 3mg/kg for those weighing between 15–35kg, and 2mg/kg for participants with a weight above 35kg. Phenobarbital was taken orally once daily with possibilities to adjust the dose based on seizure frequency and/or occurrence of side effects.”

>A priori, one has to consider two possible explanations for increase of seizures leading to need to increase the phenobarbital dose (or to change/add other antiepileptic medication, as observed in this study:

>>Pharmacokinetic interaction between ivermectine and phenobarnital leading to lowering of phenobarbital levels or free levels of phenobarbital;

>>Ivermectine induced increase of onchocerca-related breakdown products provoking seizures or lowering threshold for seizures.

>How do the authors discriminate between these two?

19. P4/L139-140: “All pregnant women were followed up until delivery”

>Were all women who became pregnant taken out from participation, and were they excluded from analyses as presented in this study?

20. P4/L141-142: “4. Primary outcome: The primary outcome was seizure freedom during the last four months of the 12-month trial”

>Here again, please specify with points in time rather than duration (or as well as duration), since duration may have a floating start in the trial time scale.

21. P4/L146-148 “5. Secondary outcomes: Skin mf density during the last four months of the trial, adverse events at any point during the trial were considered as secondary outcomes.”

>Here again, specify points in time scale.

22. Seizure monitoring

>Was seizure monitoring done blinded for any results of mf testing, including interim results?

Author Response

Response to REVIEWER #3

  1. This is a phase-4 study with two groups (1 & 2), each with two different medication regimens (Group 1, once versus thrice;  Group 2, twice versus thrice).
  2. This study started as a pilot trial of the effect of ivermectin administered once versus thrice “per” year on seizure freedom during the last four months of the 12 months trial.

>I suggest avoiding the term “per year”, since “per” inadvertently suggests a study duration lasting several years, and this wording also implicitly suggests that continued medication beyond 12 months has an effect, whereas this was not investigated.

Response: We agree to remove “per year” as suggested by the reviewer, and specify time points using “M” for month, followed by the month number of follow-up. Eg: M9.

  1. This pilot trial was registered at clinicaltrials.gov with number NCT03852303 (in this review referred to as ‘submitted protocol’); the registry indicates the trial’s status as: “completed (Actual Primary Completion Date: March, 2019; Actual Study Completion Date: July 1, 2019), with time frame for primary endpoint month 9-12, originally estimated enrollment n=150; patients included n=94; participants actually enrolled n=197; results not reported”. 

Response: We now include the results of this proof of concept trial: “A multiple logistic regression analysis showed a borderline association between ivermectin treatment and being seizure-free at month 4 (OR: 1.652, 95% CI 0.975-2.799; p=0.062) [9].” and refer to the paper describing the results of this trial.

Mandro M, S.F.J., Mukendi D, Dusabimana A, Menon S, Haesendonckx S, Lokonda R, Nakato S, Nyisi F, Abhafule G, Wonya'Rossi D, Jakwong JM, Suykerbuyk P, Meganck J, Hotterbeekx A, Colebunders R. . Ivermectin as an adjuvant to anti-epileptic treatment in persons with onchocerciasis-associated epilepsy: A randomized proof-of-concept clinical trial. PLoS Negl Trop Dis 2020, 2020 Jan 10;14(1):e0007966.

  1. The primary endpoint of the study was seizure freedom during the last four months.

>This can only be a valid end point if groups for comparison each had a similar pre-trial seizure frequency.

Response: Within each group (1&2), baseline seizure frequency were similar, hence we could conveniently compare outcomes for participants within the same group. First, we analysed the effect of ivermectin on seizure freedom in group 1 (see Table 3) and group 2 (see Table 4) adjusted for pre-trial seizure frequency.

Secondly, in the combined analysis for group 1 and group 2, we adjusted for group membership and pre-trial seizure frequency (Table 5).

>Having seizure freedom during the last four months of the trial as an endpoint, requires a comparison with a sequential series of pre-trial four months periods, for example over one pre-trial year, during which at least the number of seizure-freedom periods of four months had been monitored.

Response: The study was done in a very remote villages in the Ituri province in the DRC with security problems. Therefore there was no time to first enrol participants in a long pre-trial period. Seizure freedom during the last four months was chosen as endpoint based on previous observations in Logo, where it was noted that 80% of persons with epilepsy who had received onchocerciasis treatment became seizure-free for 4 months. This was the same approach used to estimate the sample size during the initial 4-month proof-of-concept trial of ivermectin treatment in persons with OAE (Ref: Mandro et al. Ivermectin as an adjuvant to anti-epileptic treatment in persons with onchocerciasis-associated epilepsy: A randomized proof-of-concept clinical trial. PLoS Negl Trop Dis 2020; 14: e0007966).

>Since Group 1 and Group 2 had different inclusion criteria with respect to pre-trial seizure frequency per months (less than 2 or even zero?), such comparison per individual requires not only seizure freedom per 4 months, but also seizure frequency!

Response: Zero seizures per month at baseline meant that although the participants had active epilepsy, they did not experience seizures every month (seizure frequency <12 per year).

We agree that seizure frequency is an important outcome

Baseline seizure frequency was already included as a covariate in the logistic regression models. We now included a zero-inflated Poisson regression model to investigate factors associated with seizure frequency during the last four months (results presented in Supplementary Table S3).

>In anti-epileptic drug trials, a commonly used definition of clinical effect is 50% reduction of seizures per pre-defined period in at least 50% of patients. How are results when the data are analyzed in this way, taking also into account the previous remarks under 4.? See also remarks at 5. (below)

Response: We now include 50% seizure reduction as an outcome parameter (Table 6)

Variables

OR

95% CI

P-value

Ivermectin dose twice/year vs once/year

4.469

1.250

16.620

0.026

Ivermectin dose thrice/year vs once/year

2.693

1.077

6.999

0.042

Female vs male

1.049

0.544

2.024

0.890

Age

1.006

0.970

1.045

0.759

Weight (in kg)

0.991

0.956

1.027

0.631

Seizure frequency at baseline

1.445

0.915

2.355

0.134

Duration of epilepsy (years)

1.087

0.744

1.590

0.675

Microfilarial density at baseline

0.857

0.720

1.016

0.087

Optimal vs sub-optimal AED adherence during last four months

1.431

0.734

2.799

0.305

>30mg of AED increase (between M0 and M12) vs no increase

0.250

0.122

0.496

0.000

Study group 1 vs group 2

1.634

0.609

4.454

0.344

  1. The initial set of inclusion criteria included a pre-trial seizure frequency of two or more seizures per month. (See also point 8, below)

>This inclusion criterion is mentioned in the manuscript, but NOT in the submitted protocol;

Response: The initial criteria of two or more seizures per month was mentioned in the trial protocol of the 4 month proof of concept trial (NCT03052998) but not in the protocol of 12 months trial (NCT03852303).

>This inclusion criterion is a relevant component of study design since reduction of seizures to none during the last four months of the study (month 9-12 according to submitted protocol) is the primary endpoint, and the pre-trial seizure frequency affects statistical power;

Response: We would have preferred to have more individuals with a higher number of seizures included to have more statistical power. However there were not enough individuals with at least two seizures per month in these relatively small villages.

>The length of observation of pre-trial seizures needs to be specified;

Response: There was no pre-trial follow-up. Baseline seizure frequency were established via thorough history taking with the person with epilepsy (PWE) and his/her family. At enrolment, the PWE/caretakers were asked to be as precise as possible in providing the number of seizures experienced each month during the last 3 months, and the average was recorded as baseline seizure frequency.

>Discarding the inclusion criterion of two or more pre-trial seizures per month for inclusion into Group 2 obviously would result in reduction of power to observe an effect of the trial medication.

Response: Indeed enrolling participants with fewer seizures will decrease the power of the study. Therefore we tried to increase the sample size.

>The difference in inclusion criteria between Group 1 and Group 2 – seizure frequency - directly relates to the endpoint (seizure freedom) and may have introduced significant bias. How controlled for this?

Response: First, we analysed the effect of ivermectin on seizure freedom in group 1 (see Table 3) and group 2 (see Table 4) adjusted for pre-trial seizure frequency. Secondly, in the combined analysis for group 1 and group 2, we adjusted for group membership and pre-trial seizure frequency (Table 5).

>See also remarks under 4. (above)

  1. Discrepancies in numbers between submitted protocol, figures 1 and 2, and text: Group 1: Figure 1 counts n=44 exposed once (21+23) and 46 exposed thrice (22+24), total of n=90, whereas the text (P5/Lines 196-200) counts 45 exposed once and 49 exposed thrice, total of n=94. Group 2: Figure 1 counts n=30 exposed twice and n=30 exposed thrice, whereas the text (P5/Lines 196-200) counts n=52 exposed twice and n=51 exposed thrice total of n=103.

>This needs explanation or correction, and check for impact on analysis results, if any.

Response: We adapted Figure 1. It is now in line with the protocol and the text (lines 199-200).

>It seems that the data Figure 1 represents not a flow diagram of the study as conducted, but reflects more of the originally estimated inclusion, and not the actual inclusion after ad interim adjustment of the protocol.

Response: Fig 1 is a schematic representation of the recruitment, treatment and follow-up plan of the trial. We are sorry that a pre-final version of Fig 1 was uploaded by mistake. We have now provided a final version with the correct numbers per arm.

>Further confusion arises by the text on P2/Line60, where the intended inclusion is given as n=110, a number that cannot be derived by any figure in the submitted protocol, figures or text elsewhere in this manuscript.

Response: This sample size was calculated for the 4-month proof-of-concept trial, and is well explained both in the published protocol and in the published results of this trial (Ref: Mandro et al. Ivermectin as an adjuvant to anti-epileptic treatment in persons with onchocerciasis-associated epilepsy: A randomized proof-of-concept clinical trial. PLoS Negl Trop Dis 2020; 14: e0007966).

>In the submitted protocol, the originally estimated inclusion is given as n=150. Figure 2: re 100 participants treated thrice: “24 did not receive all three doses” (P7/L220) “10 missed a dose because not visiting the health center” (P7/221)

>>Were the 10 missing a dose included in the 24 not receiving all three doses?

Response: Yes

>>Were these 24 and 10 excluded from further analysis? If not, or differentially, how was skipping one or two doses taken into account in the analysis?

Response: We performed two types of analysis: Intent-to-treat (whereby all randomized participants were included, irrespective of whether they completed follow-up) and As-treated (whereby each participant was categorized based on the actual number of ivermectin doses received).

>>What was the reason for not receiving all three doses in the 24? (Why is this specified for the 10 missing one dose, and not for the 24?)

Response: The reasons vary from withdrawal of consent, to death, or pregnancy of the participant. We now mention these reasons in the new Fig 2.

>Thrice treated group: N=100 (49 from Group 1 + 51 from Group 2) The text specifies lost to follow-up (n=12), consent withdrawn (n=6), death n=5).

>>Assuming that these three categories are mutually exclusive and subtracting the total of these three (n=23) from n=100 participants, the number of retained participants would be n=77 and not n=80.

 Response: Numbers were corrected

  1. The first group (Group 1) started as pilot study in line with the submitted protocol, but continued with randomization towards once versus thrice. The second group (Group2) comparing twice versus thrice “per” annuum was added afterwards. This was done in order to cope with the problem of low recruitment, apparent lack of statistical power, and perceived pressure from the community i.e. the population of onchocerca infected persons with epilepsy (OIPWE), to participate, likely because of benefits expected from these subsequent candidate participants.

>This may well have resulted in a more optimistic evaluation of outcome among participants in Group 2 compared to Group 1.

Response: This is unlikely, because all PWE were being followed up in the same way and they all benefitted from participation in the trial. Seizure evaluation was done for everyone in the same way, and clinicians were blind for skin snip / Ov16 results and trial regimen.

  1. It is good to note that the study design is one that compares different medication regimens, thus WITHOUT a placebo control.

Response: We agree, this has now been mentioned.

>In the discussion section, the authors state that the negative results of a previous study of effect of ivermectin on seizure frequency can be explained by the lack of a placebo control in that study.

Response: Our study was a comparative study with different treatment arms that included effective anti-epileptic drugs. This is very different with the study that enrolled a small number of persons with refractory (i.e. treatment-resistant) epilepsy. In the latter study there was no comparative group. Given the fact that the frequency of seizures vary it is difficult to interpret the results of this study.

>In fact, also the current study lacks placebo controls, to be stated as such in this manuscript and not to be contrasted with a previously published study with negative outcome;

Response: We now state “A small study conducted in a non-onchocerciasis endemic area reported that treatment with ivermectin decreased the frequency of seizures in persons with refractory epilepsy [21]. However in that study, ivermectin was given frequently (10mg/day) and potentially interacted with the AED metabolism of the participants. Also, the frequency of seizures may still vary considerably even among persons with refractory epilepsy. Therefore without a comparative group it is difficult to interpret the results of that study.”

>Furthermore, participants were aware of differences in medication regimens (once versus twice versus thrice);

Response: These differences were not very obvious for the participants. All participants reported to the health centre monthly for clinical follow-up and refill of anti-epileptic drugs. During these visits, they also received ivermectin according to the randomization schedule. This was also done at the health centre (directly observed). In our opinion, the fact that ivermectin treatment was provided in the context of a routine visit (not in the form of a multi-day home treatment), and that all participants had this same experience, at least once, limits the chances of a major placebo effect.

>In this respect, Group 2 is incomparable to Group 1 because of the pressure from that population to ‘benefit’ from participation, and the loosening of the inclusion criteria.

>Both may have induced a significant placebo effect.

Response: We do not understand this comment both groups benefited in the same way and were followed up in the same way. Moreover we are not comparing in this study Group 2 versus Group 1 individuals. We only are comparing persons who were randomised in different treatment arms.

  1. There is no mention of a possible role of pharmacokinetic interaction between ivermectin and phenobarbital (literature!).

>These interactions may be quite complex and perturb the interpretation of results.

>This is especially relevant since seizure frequency is the primary outcome measure and may be influenced also by phenobarbital levels.

Response: We agree with the reviewer that there may be interactions between ivermectin and phenobarbital. For this reason we used seizure freedom during the “last 4 months” as our outcome. The half-life of ivermectin being 16-18 hours, we do not expect that potential interactions with anti-epileptic drugs would last for several months. However, the effects of ivermectin on microfilariae (mf) can last for several months, and that is what this study showed: reduced mf after several months of ivermectin treatment were associated with reduced seizures.

>Effect of ivermectin on seizure frequency by effect on phenobarbital levels (induction of metabolism) can only be excluded if this has been sorted out in previous studies, or by plasma level monitoring of phenobarbital during the current study.

Response: Please see response above. This point would be only relevant if ivermectin was given daily together with anti-epileptic drugs as in the study by Diazgranados-Sanchez et al. By the end of our initial 4-month trial, phenobarbital levels were optimal among participants who had received ivermectin four months earlier and those who had not (See Table 3 in Ref: Mandro et al. PLoS Negl Trop Dis 2020; 14: e0007966). This strongly suggests that any possible effects of a single dose of ivermectin on phenobarbital metabolism should be very short-lived, and would certainly not influence phenobarbital serum levels 4 months later.

>If none is available, that would need caveat remarks in the discussion and conclusion.

>Vice versa, phenobarbital may have a negative effect on e.g. gut mucosa PGP/MDR1 mediated bioavailability of ivermectin and accelerate the catabolism of ivermectin by induction of CYP3A. How did the investigating team take these possible interactions into account while setting up the study and interpreting the data as obtained?

Response: We now mentioned this mechanism in our discussion (Ref: Ballent M, et al. Modulation of the P-glycoprotein-mediated intestinal secretion of ivermectin: in vitro and in vivo assessments. Drug Metab Dispos. 2006; 34(3):457–63. DOI: 10.1124/dmd.105.007757). If such a mechanism played a role, it would have impacted all study arms. In the presence of such a mechanism the benefit of the multiple dose ivermectin regimen over the one dose ivermectin regimen would be smaller.

  1. The inclusion criteria related to seizures/epilepsy is markedly different between Group 1 and Group 2. (see also point 4). Apparently for both group 1 and 2, the ILAE definition is used, of at least two seizure separated more than 24 hours (Quota from Page 3, lines 85-88: “To be enrolled, participants had to meet the 2014 International League Against Epilepsy (ILAE) definition of epilepsy: having experienced at least two seizures, unprovoked and without fever, with a minimal time difference of 24 hours between the two events”), thus without setting a minimum per month.

>Only for the initial pilot study (n=94), from which group 1 evolved, there was an additional inclusion criterion of at least two seizures per month. This makes Group 1 and Group 2 incomparable with respect to seizure outcome measures and statistical power calculation. For both group 1 and 2, more data is needed concerning length of pre-trial observation of seizures and calculation of seizure frequency.

Response: There was no pre-trial observation period in this study (Please see responses to point 4 and 5 above).

  1. The inclusion criteria are not consistent.

>See legend to figure 2: “OIPWE not meeting the OAE criteria“or” < 2 seizures/month at baseline”. Thus, according to figure 2, subjects were included that not had onchocerca-associated epilepsy. Why then, were they included? Or change the wording?

Response: Although some participants did not meet the OAE criteria, we included them in the study because they had onchocerciasis. Given that we had previously documented a correlation between mf density and seizure frequency (Ref: Siewe Fodjo JN, et al. Onchocerciasis-associated epilepsy in the Democratic Republic of Congo: Clinical description and relationship with microfilarial density. PLoS Negl Trop Dis 2019; 13: e0007300), we expected to witness a seizure reduction in these participants depending on the number of doses of ivermectin they will receive.

  1. Skin snip positivity and/or [antibody] Ov16 positivity (see methods as well as table 1).

>This implies that patients were regarded as having active onchocerca, whereas only antibodies were positive. Is this correct or, what is the justification for this? Please specify numbers for each of the four combination of the two (Skin snip pos/neg & ab-OV16 pos/neg)(+/+, +/-, -/+, -/-).

Response: Indeed it is considered that only persons with positive skin snips have active onchocerciasis. We also included persons with Ov16 antibodies because in an ivermectin naïve population, it is likely that these persons have an active O. volvulus even without detectable microfilariadermia (the adult worm can live for up to 15 years in the absence of treatment, and mf in skin are often mobile and could be missed by only performing two skin snips). Moreover there was an interest in enrolling individuals with different degrees of O. volvulus infection in order to show that the effect of ivermectin on the frequency of seizures was related to its anti-parasitic effect and not as an anti-epileptic drug.

See below the results of the OV16 and skin snip results at baseline 

Group 1

Skin snip positivity

OV 16 positivity

Negative

Positive

Negative

0

21

Positive

26

47

Group 2

Negative

Positive

Negative

0

34

Positive

26

43

Another paper is being written about the relationship between OV16 positivity, skin snip results and urinary NATOG.

  1. As expected, group 2 patients had less seizures and in general less disease related features at inclusion.

>In this respect, also the difference in pregnancy rate is not surprising.

Response: We agree.

  1. In Table 2 evaluation data are given at month 12. According to the scheme in Figure 1, the thrice medicated group consisted of three subgroup. For one of the three subgroups (Group 1, pilot trial), this evaluation was done three months after the third dosing of ivermectin, whereas for two of the three subgroups (Group 1, after re-randomization; Group 2, one arm) this was done three months after the second dosing, at or around the time of the third dosing.

>If so, please specify whether the evaluation in the latter was shortly before or how long after the third dose.

>If not, please specify whether evaluation at 12 months is related to point of time of entry into the study OR the start of medication.

>The last latter be the only appropriate comparison (equal duration of treatment in relation to time of evaluation).

Response: The timing in our study starts upon receiving the first dose of ivermectin (this is T=0 for each participant). Although participants in different arms received their first ivermectin at different times, they were all followed up for 12 months from that time, with some of them receiving additional doses of ivermectin at four month intervals.

>For clarity, throughout the manuscript, it is better to use consistently Preset points in time (e.g. T=0, T=4, T=8, etc) than to mix points in times with durations. In this way, durations (of exposure) can be derived uniformly.

Response: This has now been done. We propose to use the notation M0, M4, M8, etc. (M for “Month”)

  1. Was the skin snip done always shortly before each dosing of ivermectin (appropriate), afterwards, or irregularly, or otherwise?

Response: Skin snips were obtained during the follow-up visits at T=0, 4, 8, 12. Always done before treatment with ivermectin.

  1. Figure 3. At baseline and at 8 and 12 months of the once per year treatment arm, Group 1 (A) was not different from that of the twice and thrice arm, Group 2 (B).

>This leaves the question unanswered whether there is a significant difference at 4 months and, if so, how to interpret its clinical significance.

Response: The outcomes at T=4 have been extensively discussed in a previous publication (Ref: Mandro et al. PLoS Negl Trop Dis 2020; 14: e0007966). A multiple logistic regression analysis showed a borderline association between ivermectin treatment and being seizure-free at month 4 (OR: 1.652, 95% CI 0.975-2.799; p=0.062). There was a significant decrease in mf density among participants who had received ivermectin at T=0 compared to those who were only treated with ivermectin at T=4.

  1. P4/L109-110: “The minimal time interval between two consecutive doses of ivermectin was four months.”

>What was, in practice during this study, the variation in time interval?

>What was the maximum interval between two successive doses allowed for continuation of study participation?

>Was there a difference between study-groups and -arms that may have had an effect on interpretation of the results?

Response: The time interval between two doses was ALWAYS 4 months (see Figure 1). The word “minimal” was a mistake, which has now been corrected in the manuscript.

  1. P4/L112-116: “The AED (phenobarbital) dose was based on the participants’ weight: 5mg/kg for participants weighing <15kg; 3mg/kg for those weighing between 15–35kg, and 2mg/kg for participants with a weight above 35kg. Phenobarbital was taken orally once daily with possibilities to adjust the dose based on seizure frequency and/or occurrence of side effects.”

>A priori, one has to consider two possible explanations for increase of seizures leading to need to increase the phenobarbital dose (or to change/add other antiepileptic medication, as observed in this study:

>>Pharmacokinetic interaction between ivermectine and phenobarnital leading to lowering of phenobarbital levels or free levels of phenobarbital;

Response: This is unlikely, as the half-life of ivermectin is only 18 hours. Meanwhile, daily intake of phenobarbital would cause the drug to reach a steady state within just two weeks and its effects on seizure control would be evident during the monthly follow-up visit.

>>Ivermectine induced increase of onchocerca-related breakdown products provoking seizures or lowering threshold for seizures.

Response: Ivermectin in the 4 month proof of concept trial was shown not to increase seizures and most likely reduced seizures in onchocerciasis-infected epileptic patients. Indeed, a multiple logistic regression analysis showed a borderline association between ivermectin treatment and being seizure-free at month 4 (OR: 1.652, 95% CI 0.975-2.799; p=0.062). (Ref: Mandro et al. PLoS Negl Trop Dis 2020; 14: e0007966).

>How do the authors discriminate between these two?

Response: There is no evidence from the 4 month proof of concept trial nor from this 12 month trial that treatment with ivermectin lowers the threshold for seizures.

  1. P4/L139-140: “All pregnant women were followed up until delivery”

>Were all women who became pregnant taken out from participation, and were they excluded from analyses as presented in this study?

Response: Pregnant women were kept in the study to monitor monthly seizure frequency and mf density in skin snips every 4 months. However, no further doses of ivermectin were given to them after the pregnancy had been diagnosed. This has been mentioned in the manuscript.

  1. P4/L141-142: “4. Primary outcome: The primary outcome was seizure freedom during the last four months of the 12-month trial”

>Here again, please specify with points in time rather than duration (or as well as duration), since duration may have a floating start in the trial time scale.

Response: We now mention: “seizure freedom between M9 and M12 of follow-up.”

  1. P4/L146-148 “5. Secondary outcomes: Skin mf density during the last four months of the trial, adverse events at any point during the trial were considered as secondary outcomes.”

>Here again, specify points in time scale.

Response: We now mention: “Skin mf density between M9 and M12 of follow-up.”

  1. Seizure monitoring

>Was seizure monitoring done blinded for any results of mf testing, including interim results?

Response: Yes. The clinical staff involved in monitoring seizure frequency were always blind for ivermectin treatment regimen and skin snip results.

Round 2

Reviewer 1 Report

The authors have responded to my concerns.

Reviewer 3 Report

Given the nature of the study (settings, design, population), the authors have responded well to the reviewer's remarks and suggestions.